# Long-term risk of autoimmune diseases after mRNA-based SARS-CoV2 vaccination in a Korean, nationwide, population-based cohort study

Seung-Won Jung [1,2], Jae Joon Jeon [1,2], You Hyun Kim [1], Sung Jay Choe [1] ✉ & Solam Lee [1] ✉

The long-term association between mRNA-based coronavirus disease 2019 (COVID-19) vaccination and the development of autoimmune connective tissue diseases (AI-CTDs) remains unclear. In this nationwide, population-based cohort study involving 9,258,803 individuals, we aim to determine whether the incidence of AI-CTDs is associated with mRNA vaccination. The study spans over 1 year of observation and further analyses the risk of AI-CTDs by stratifying demographics and vaccination profiles and treating booster vaccination as time-varying covariate. We report that the risk of developing most AI-CTDs did not increase following mRNA vaccination, except for systemic lupus erythematosus with a 1.16-fold risk in vaccinated individuals relative to controls. Comparable results were reported in the stratified analyses for age, sex, mRNA vaccine type, and prior history of non-mRNA vaccination. However, a booster vaccination was associated with an increased risk of some AI-CTDs including alopecia areata, psoriasis, and rheumatoid arthritis. Overall, we conclude that mRNA-based vaccinations are not associated with an increased risk of most AI-CTDs, although further research is needed regarding its potential association with certain conditions.

Infection with severe acute respiratory syndrome coronavirus 2 (SARS-CoV-2) results in the development of coronavirus disease 2019 (COVID-19), and it has spread globally since 2020. COVID-19 has emerged as a notable pandemic, causing a substantial burden on public health, as >50% of the world's population has been infected with it, according to a 2022 global seroprevalence survey[1,2].

Shortly after the COVID-19 outbreak, vaccines emerged as a crucial intervention to address the pandemic. These vaccines were developed primarily by two main technological platforms replicating incompetent adenoviral vectors and mRNA[3]. In particular, though the effectiveness of vaccine wanes over time and as virus variants such as Omicron, mRNA-based COVID-19 vaccines show generally significant

efficacy, preventing 46–92% of SARS-CoV-2 infections, 74–87% of hospitalisations, and 62–92% of severe illnesses, as defined by the National Institutes of Health criteria[4]. Moreover, recent clinical research or systematic reviews concerning mRNA COVID-19 vaccines have confirmed their generally favourable safety profiles[5–7].

Although the COVID-19 vaccine has played a crucial role in combatting the pandemic, the mRNA COVID-19 vaccine is reportedly also associated with adverse events, notably cardiac complications such as myo- and pericarditis[8–10]. In particular, the potential association between the mRNA COVID-19 vaccine and autoimmune connective tissue diseases (AI-CTDs) has been actively studied. A few systemic autoimmune diseases, such as autoimmune hepatitis and

---

[1]Department of Dermatology, Yonsei University Wonju College of Medicine, Wonju, Republic of Korea. [2]These authors contributed equally: Seung-Won Jung, Jae Joon Jeon. ✉e-mail: wow8561@yonsei.ac.kr; solam@yonsei.ac.kr

nephropathies, have been identified as being potentially associated with mRNA vaccines[11,12]. While previous studies have suggested an association between mRNA vaccines and several systemic auto-immune diseases, there are limited studies demonstrating the development of AI-CTDs following mRNA vaccination in large populations over a period of >1 year, despite the low incidence and slow development of AI-CTDs. These uncertainties and adverse effects of mRNA vaccines have heightened public scepticism regarding vaccination and necessitated a risk-benefit analysis of vaccination.

In this study, we aim to determine whether the incidence of AI-CTDs is associated with mRNA vaccination against SARS-CoV2.

## Results
### Study population
The primary cohort was established by combining the National Health Insurance Service (NHIS) and Korea Disease Control and Prevention Agency (KDCA) databases, which comprised the healthcare data of > 99% of the entire Korean population and their COVID-19 diagnosis

and vaccination profiles. In total, 9,258,803 individuals who had received at least one dose of the mRNA-based COVID-19 vaccine were included (Fig. 1). As COVID-19 vaccines have been administered nationwide in South Korea, the proportion of unvaccinated individuals was minimal[13]. Therefore, the use of unvaccinated individuals as comparators could have resulted in improper cohort selection and potential selection bias. We consequently established a historical control cohort within mRNA-vaccinated individuals, but the observational period was shifted back 2 years from the date of the first dose of mRNA vaccination of the historical control cohort. In total, 4,445,333 and 4,444,932 patients were included in the vaccination and historical control cohorts, respectively, and all were observed for ≥1 year. The baseline demographic and general health characteristics of each cohort are summarised in Table 1. The covariates were well-balanced after the inverse probability of treatment weighting (IPTW). The COVID-19 vaccination profiles, such as the type of mRNA vaccine or history of non-mRNA vaccination, are summarised in Supplementary Table 1. The mean follow-up times for the vaccination and historical

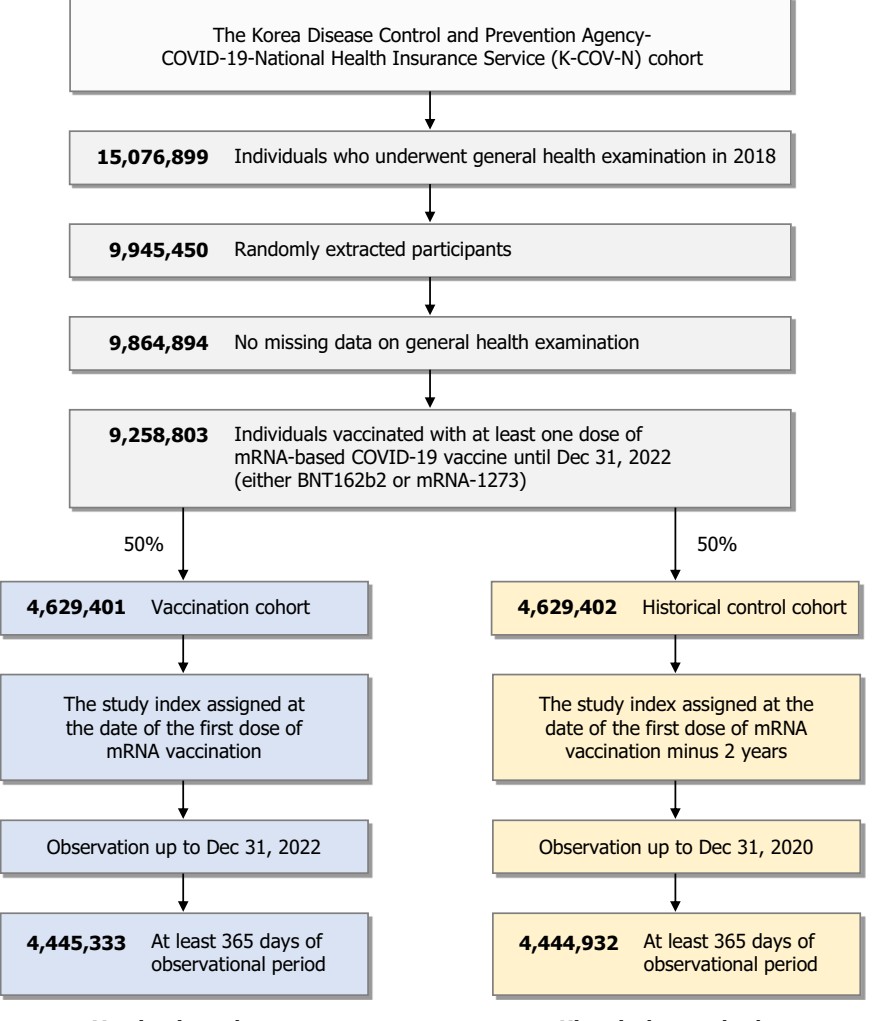

**Fig. 1 | Flowchart of study population selection.** This nationwide population-based cohort study combined data from the Korea Disease Control and Prevention Agency (KDCA) and the COVID-19 National Health Insurance Service (NHIS) cohort (K-COV-N cohort). The study included approximately 20% of the total South Korean population. This primary cohort comprised all individuals vaccinated with at least one dose of the mRNA-based COVID-19 vaccine (BNT162b2, Pfizer-BioNTech; mRNA-1273, Moderna) until 31 December 2022. Subsequently, half of the primary cohort was extracted to establish the vaccination cohort, and the study index was

defined as the date of the first dose of the mRNA-based COVID-19 vaccine. A historical control cohort was established by extracting the other half of the primary cohort as the control, and its study index was assigned as the date of the first dose of the mRNA-based COVID-19 vaccine minus 2 years. A total of 4,445,333 vaccination and 4,444,932 control cohorts were selected and observed until 31 December 2022 and 31 December 2020, respectively. Abbreviations: COVID-19, Coronavirus 2019 disease.

**Table 1 | Demographic and general health characteristics of the COVID-19 vaccination cohort and the historical control cohort before and after inverse probability treatment weighting**

| Characteristics | Preweighting, patients, No. (%) | | | Postweighting, weighted (%) | | |
|---|---|---|---|---|---|---|
| | COVID-19 vaccination (N = 4,445,333) | Historical control (N = 4,444,932) | SMD | COVID-19 vaccination (N = 4,445,333) | Historical control (N = 4,444,932) | SMD |
| Age, mean, ± SD, y | 53.42 ± 14.19 | 51.42 ± 14.19 | 0.141 | 52.42 (20.03) | 52.41 (20.11) | 0.001 |
| < 40, n (%) | 1,773,871 (39.91) | 2,010,298 (45.22) | | (42.86) | (42.26) | – |
| 40-59, n (%) | 2,021,790 (45.48) | 1,908,587 (42.94) | | (43.96) | (44.56) | |
| > 60, n (%) | 649,672 (14.61) | 526,047 (11.84) | | (13.18) | (13.18) | |
| Sex, n (%) | – | – | 0.000 | – | – | 0.000 |
| Male | 2,384,742 (53.65) | 2,383,958 (53.63) | | (53.63) | (53.63) | – |
| Female | 2,060,591 (46.35) | 2,060,974 (46.37) | | (46.37) | (46.37) | |
| Insurance type, n (%) | – | – | 0.002 | – | – | 0.000 |
| Standard | 4,369,010 (98.28) | 4,367,357 (98.25) | | (98.27) | (98.27) | – |
| Medicaid | 76,323 (1.72) | 77,575 (1.75) | | (1.73) | (1.73) | |
| Income level quartile, n (%)[a] | – | – | 0.001 | – | | 0.000 |
| Highest | 705,154 (15.84) | 706,185 (15.89) | | (15.53) | (16.19) | – |
| Higher | 1,078,272 (24.26) | 1,079,213 (24.28) | | (24.57) | (23.99) | |
| Lower | 1,262,642 (28.40) | 1,259,636 (28.34) | | (28.79) | (27.96) | |
| Lowest | 1,400,265 (31.50) | 1,399,903 (31.49) | | (31.10) | (31.85) | |
| Area of residence, n (%) | – | – | 0.001 | – | – | 0.000 |
| Urban area | 1,932,306 (43.47) | 1,933,980 (43.51) | | (43.49) | (43.49) | – |
| Rural area | 2,513,027 (56.53) | 2,510,952 (56.49) | | (56.51) | (56.51) | |
| Underlying disease, n (%) | | | | | | |
| Hypertension | 1,093,732 (24.61) | 1,283,097 (28.86) | 0.096 | (26.70) | (26.68) | 0.000 |
| Diabetes mellitus | 700,779 (15.76) | 542,836 (12.21) | 0.103 | (13.97) | (13.96) | 0.000 |
| Dyslipidemia | 1,806,828 (40.65) | 1,407,281 (31.66) | 0.189 | (36.15) | (36.14) | 0.000 |
| Atopic dermatitis | 38,558 (0.87) | 27,588 (0.62) | 0.029 | (0.74) | (0.75) | 0.000 |
| Allergic rhinitis | 201,939 (4.54) | 176,682 (3.97) | 0.028 | (4.26) | (4.26) | 0.000 |
| Asthma | 77,511 (1.74) | 68,092 (1.53) | 0.017 | (1.64) | (1.64) | 0.000 |
| Hypothyroidism | 185,241 (4.17) | 136,800 (3.08) | 0.058 | (3.62) | (3.62) | 0.000 |
| Hyperthyroidism | 66,441 (1.49) | 49,313 (1.11) | 0.034 | (1.30) | (1.30) | 0.000 |
| Hashimoto thyroiditis | 29,350 (0.66) | 21,710 (0.49) | 0.023 | (0.58) | (0.58) | 0.000 |
| Vitamin D deficiency | 178,225 (4.01) | 94,993 (2.14) | 0.109 | (3.07) | (3.07) | 0.000 |
| Hepatitis B | 96,388 (2.17) | 81,790 (1.84) | 0.023 | (2.00) | (2.00) | 0.000 |
| Hepatitis C | 12,079 (0.27) | 10,298 (0.23) | 0.008 | (0.25) | (0.25) | 0.000 |
| HIV infection | 163 (0.00) | 143 (0.00) | 0.001 | (0.00) | (0.00) | 0.000 |
| General health examination data, n (%) | | | | | | |
| Current smoker | 851,737 (19.16) | 852,265 (19.17) | 0.000 | (19.16) | (19.16) | 0.000 |
| Drinking[c] | 2,840,913 (63.91) | 2,839,130 (63.87) | 0.001 | (63.86) | (63.87) | 0.000 |

[a]The income level was divided into quartiles based on health insurance premiums.
[b]Body mass index was calculated as weight in kilograms divided by height in metres squared.
[c]Drinking was defined as routine alcohol consumption, regardless of the amount or frequency.
Abbreviations: COVID-19, Coronavirus 2019 disease; HIV, human immunodeficiency virus; SMD, absolute standardised mean difference.

control cohorts were 471.24 ± 66.16 days and 471.28 ± 66.15 days, respectively.

**Autoimmune connective tissue diseases following mRNA vaccination**

Cumulative incidence plots for the AI-CTDs are shown in Fig. 2, with Supplementary Fig. 1 providing additional details, including cumulative incidence for positive and negative control outcomes, as well as the cumulative number of events for each time point. The risks of developing incident AI-CTDs in the vaccination and historical control cohorts are shown in Fig. 3. To mitigate the risk of type I error induced by multiple comparisons, we employed a Bonferroni correction for 27 predefined outcomes and used an adjusted 99.81% confidence interval (99% CI) to determine statistical significance. Individuals who had the

mRNA COVID-19 vaccine did not incur higher risks of developing most AI-CTDs such as alopecia areata (adjusted hazard ratio [aHR], 1.00; 99% CI, 0.96–1.04), alopecia totalis (aHR, 0.79; 99% CI, 0.68–0.93), psoriasis (aHR, 0.80; 99% CI, 0.77–0.84), vitiligo (aHR, 0.95; 99% CI, 0.88–1.02), anti-neutrophil cytoplasmic antibody (ANCA) associated vasculitis (aHR, 1.09; 99% CI, 0.72–1.66), sarcoidosis (aHR, 1.06; 99% CI, 0.78–1.44), Behcet disease (aHR, 0.69; 99% CI, 0.58–0.82), Crohn's disease (aHR, 0.92; 99% CI, 0.77–1.09), ulcerative colitis (aHR, 0.97; 99% CI, 0.87–1.08), rheumatoid arthritis (aHR, 0.86; 99% CI, 0.84–0.89), systemic sclerosis (aHR, 1.01; 99% CI, 0.73–1.38), Sjogren's syndrome (aHR, 1.07; 99% CI, 0.96–1.18), ankylosing spondylitis (aHR, 0.95; 99% CI, 0.87–1.04), dermato/polymyositis (aHR, 1.02; 99% CI, 0.77–1.35), and bullous pemphigoid (BP) (aHR, 1.53; 99% CI, 0.90–2.60). However, individuals in the mRNA vaccination cohort

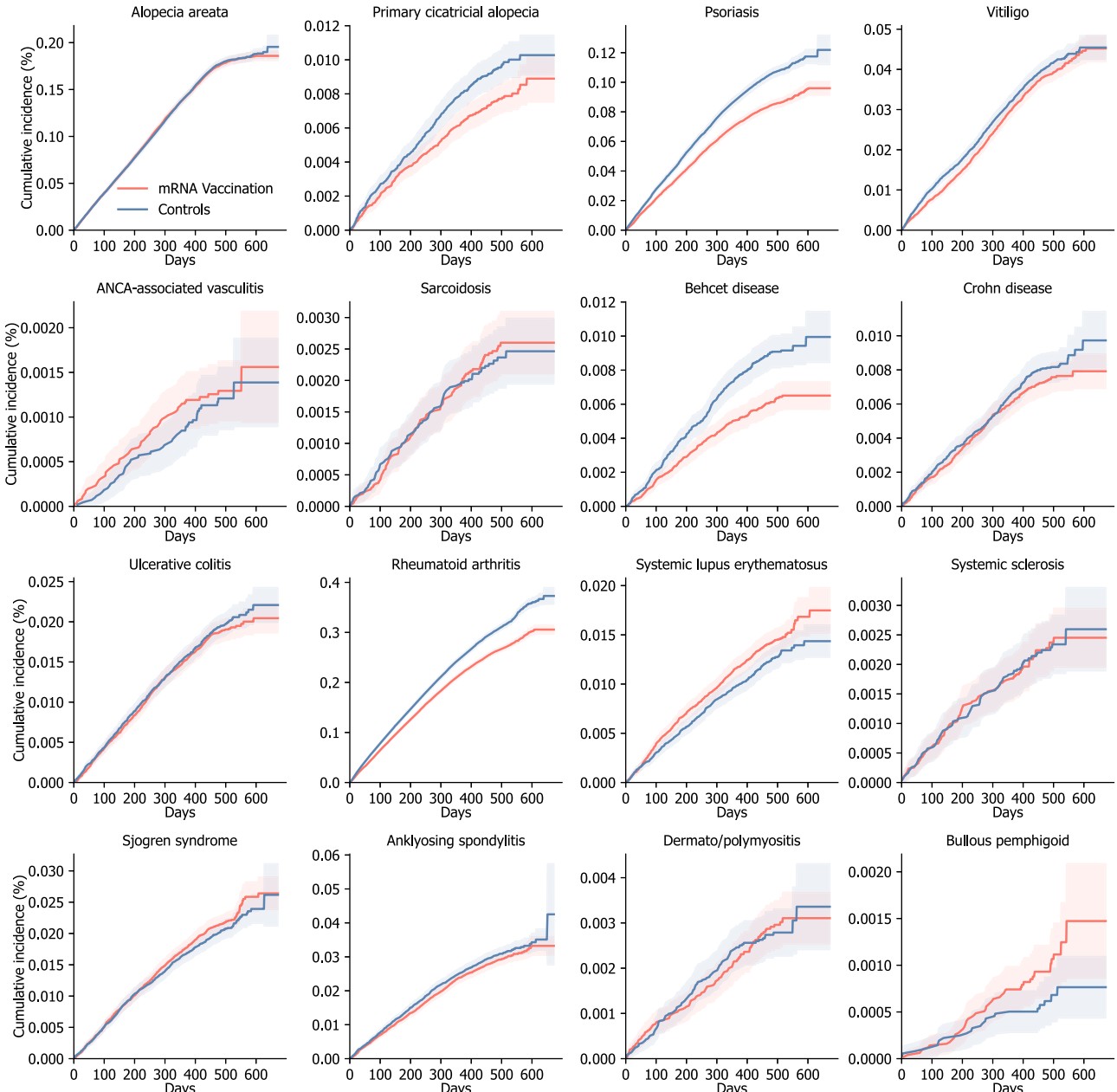

**Fig. 2 | Cumulative incidences of autoimmune connective tissue diseases outcomes.** The cumulative incidence plot shows the cumulative incidences of autoimmune connective tissue diseases in mRNA-based COVID-19 vaccination cohort and historical control cohort. The shaded area shows a 95% confidence interval for the cumulative incidences. Additional information, including cumulative incidence for positive and negative control outcomes, as well as the cumulative number of events for each time point, was presented in Supplementary Fig. 1. Abbreviation: ANCA, anti-neutrophil cytoplasmic antibody; COVID-19, coronavirus disease 2019.

were at considerably higher risk of developing systemic lupus erythematosus (SLE) (aHR, 1.16; 99% CI, 1.02–1.32) than those in the historical control cohort.

## Validation of the results using positive and negative control outcomes

To validate these findings, we evaluated the risks of positive and negative control outcomes associated with mRNA vaccination. For the positive control outcomes, the risk of myocarditis (aHR, 7.20; 99% CI, 4.37–11.86), pericarditis (aHR, 2.75; 99% CI, 1.95–3.88), and Guillain–Barre syndrome (aHR, 1.62; 99% CI, 1.16–2.25) were considerably higher in the vaccination cohort than in the historical control cohort (Fig. 3). Conversely, the risk of having negative control outcomes was not considerably higher in the vaccination cohort than in the historical control cohort (benign skin tumour (aHR, 1.02; 99% CI,

1.00–1.05), melanoma in situ (aHR, 1.21; 99% CI, 0.64–2.29), and tympanic membrane perforation (aHR, 0.84; 99% CI, 0.77–0.91)).

## Stratified analysis by sex, age, and vaccination profile

In subgroup analyses, we compared the vaccination and historical control cohorts stratified by sex, age (< 40 vs. ≥ 40), type of mRNA-based COVID-19 vaccine (BNT162b2, Pfizer–BioNTech vs. mRNA-1273, Moderna), cross-vaccination status with any history of non-mRNA COVID vaccination (ChAdOx1 nCoV-19 [AZD1222], Oxford–AstraZeneca or Ad26.COV2.S, Janssen–Johnson & Johnson, or others) prior to mRNA vaccination, and any history of COVID-19 diagnosis. In general, there were no significant differences between the two cohorts in the subgroup analyses for most outcomes (Figs. 4–5), with Supplementary Figs. 2–11 providing additional details for each stratified analysis. However, women who had received the mRNA vaccine had a significantly higher risk of developing BP (aHR, 2.67;

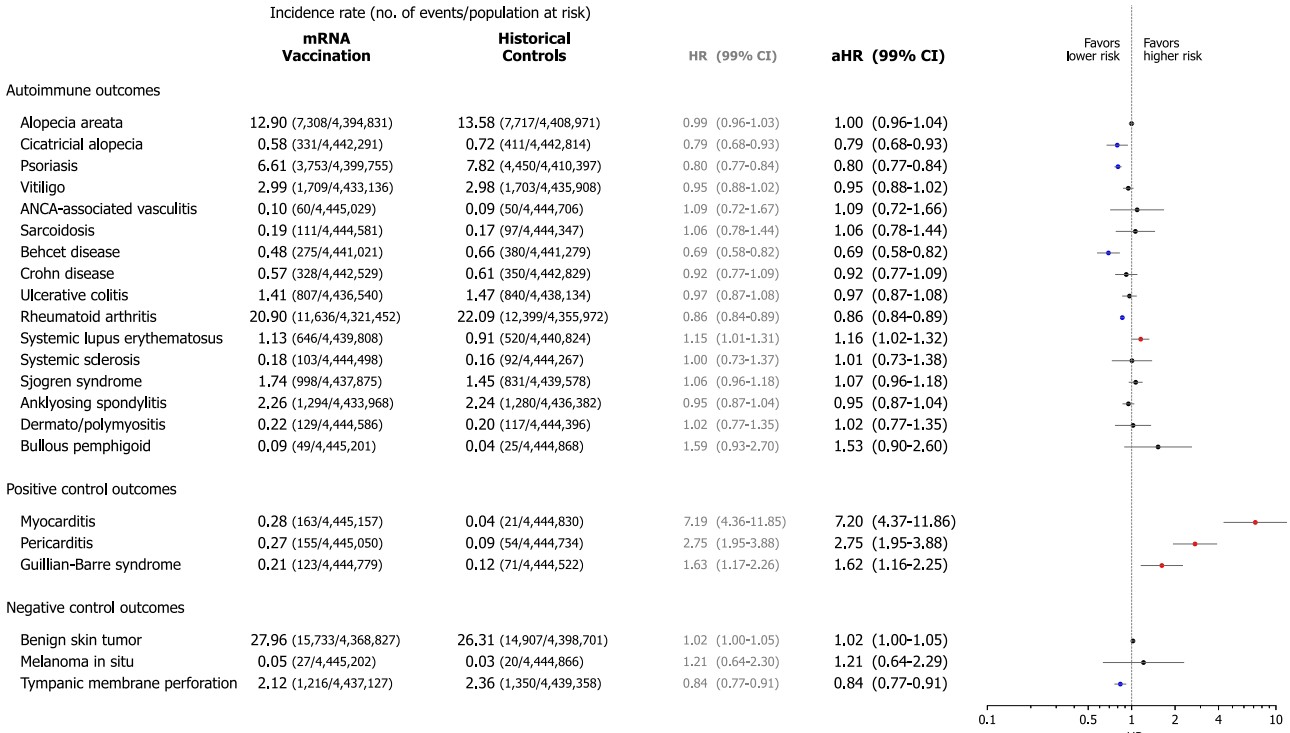

**Fig. 3 | Risks of incident autoimmune skin and connective tissue disorders in the mRNA-based COVID-19 vaccination cohort compared with the historical control cohort.** To minimise the differences in baseline characteristics between the vaccination and historical control cohorts, predefined covariates, including demographics, socioeconomic status, and comorbidities, were balanced using inverse probability of treatment weighting. Subsequently, the incidence in the vaccination cohort compared to that in the historical control cohort was estimated using multivariable Cox proportional hazards analysis after adjusting for all pre-defined covariates. The forest plot depicts adjusted hazard ratios (aHRs) in individuals with mRNA-based COVID-19 vaccination compared with historical controls, with the confidence interval (CI) adjusted to 99.81% for Bonferroni correction but presented as 99% CI for simplicity. The point estimate (centre) represents the aHR, and the horizontal line (error bar) shows the range of the 99% CI. The incidence rate was calculated as the number of events divided by 10,000 person-years, with the population at risk also presented. Abbreviations: aHR, adjusted hazard ratio; ANCA, antineutrophil cytoplasmic antibody; CI, confidence interval; COVID-19, coronavirus disease 2019; HR, hazard ratio.

99% CI, 1.11–6.42) (Fig. 4a, b). In addition, aged ≥40 years who had undergone mRNA vaccination tended to have a higher risk of developing BP (aHR, 1.53; 99% CI, 0.90–2.61) (Fig. 4c, d). In the stratified analysis based on the type of mRNA vaccine received, individuals who received the BNT162b2 vaccine had a significantly higher risk of developing SLE (aHR, 1.18; 99% CI, 1.02–1.36) (Fig. 5a, b). In addition, cross-vaccination with non-mRNA vaccines did not independently affect the incident risk of any AI-CTDs (Fig. 5c, d). In analyses according to the status of COVID-19 diagnosis, the incidence was not higher for almost all AI-CTDs, except SLE in individuals with COVID-19 diagnosis (aHR, 1.23; 99% CI, 1.05–1.44) (Supplementary Figs. 6 and 7).

## Booster vaccination

In total, 2,284,342 individuals had the booster mRNA vaccination (3rd dose of mRNA vaccination) among the vaccination cohort. In extended Cox proportional hazard analyses treating booster vaccination as time-varying covariate, the risk of alopecia areata (aHR, 1.12; 99% CI, 1.05–1.19), psoriasis (aHR, 1.16; 99% CI, 1.06–1.27), and rheumatoid arthritis (aHR, 1.14; 99% CI, 1.08–1.21) were greater in individuals who had booster vaccination compared to those who had not (Fig. 6).

## Discussion

BNT162b2 and mRNA-1273 were the first mRNA vaccines approved by the US Food and Drug Administration (FDA) for combatting COVID-19[14]. While prior studies have suggested that non-mRNA vaccinations or COVID-19 infections increase the risk of patients developing autoimmune diseases, data on the long-term effects of mRNA vaccine administration on AI-CTDs are scarce[15–17]. Our study investigated the effect of mRNA vaccination on the occurrence of AI-CTDs by observing at least 1-year period in a nationwide population-based setting comprising more than 8 million individuals. In this analysis, the incidence of most AI-CTDs was not associated with mRNA vaccination. However, we observed an increased risk of developing some AI-CTDs after booster vaccination.

We previously reported no significant difference in the risk of developing AI-CTDs between the mRNA vaccination group and the historical control group at a mean follow-up of 100 days[18]. Our results were generally aligned with the previous study, but we found some gaps in an increased risk of some AI-CTDs, including SLE. This may be attributed to differences in the demographic characteristics of the study population and observational periods across the studies. Given the indolent course of AI-CTDs, these results suggest that long-term surveillance for the development of AI-CTDs after mRNA vaccination may be warranted.

Although the association between mRNA vaccination and SLE remains unclear, there have been cases in which SLE has developed following mRNA vaccination[19]. mRNA vaccination reportedly leads to elevated plasma anti-dsDNA antibody levels, and the extracellular self-DNA influences the pathogenesis of AI-CTDs, including SLE[20,21]. Another study found that booster vaccinations increase circulating cell-free DNA in B cells, T cells, and monocytes[22]. In addition, the observed risk of SLE varied according to the type of mRNA vaccine and the history of non-mRNA vaccination. Further studies are needed to elucidate whether factors such as mRNA dose may contribute to these differences[23,24].

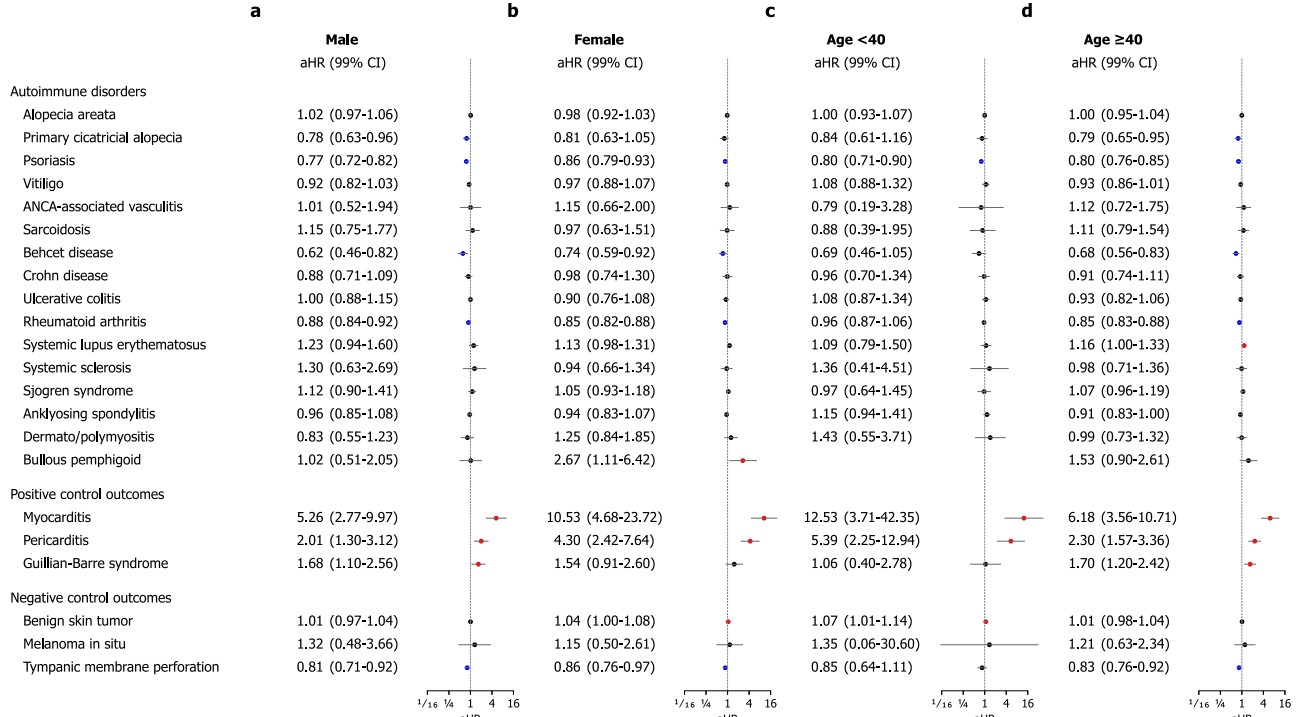

**Fig. 4 | Stratified analyses of the risks of incident autoimmune connective tissue disorders in the mRNA-based COVID-19 vaccination cohort compared with the historical control cohort according to sex and age.** The forest plot depicts adjusted hazard ratios (aHRs) and 99% confidence intervals (CIs) in individuals with mRNA-based COVID-19 vaccination compared with historical controls. The point estimate (centre) represents the aHR, and the horizontal line (error bar) shows the range of the 99% CI. The incident risks of autoimmune disorder outcomes were stratified by sex ((**a**) Male or (**b**) Female) and age ((**c**) < 40 years or (**d**) ≥ 40 years). Additional details, including unadjusted HRs and population at risk, were provided in Supplementary Figures.; Male subgroup (Supplementary Fig. 2), female subgroup (Supplementary Fig. 3), subgroup aged < 40 years (Supplementary Fig. 4), and subgroup aged ≥ 40 years (Supplementary Fig. 5). Abbreviations: aHR, adjusted hazard ratio; ANCA, antineutrophil cytoplasmic antibody; CI, confidence interval; COVID-19, coronavirus disease 2019.

The association between BP and mRNA vaccination remains to be elucidated, however, the vaccinated female population shows a 2.67-fold higher risk of BP development than the non-vaccinated female population in the subgroup analysis stratified by sex. The US case series of subepidermal blistering eruptions (including BP) following mRNA vaccination reported it to be more common in women and after the age of 40[25]. Similarly, our study tended to a higher risk of BP in women and in patients over 40 years of age following mRNA vaccination than in historical controls. This result may suggest the need to monitor BP development in females who have received mRNA-based vaccines.

Furthermore, our study found that booster vaccination was associated with an increased risk of developing certain AI-CTDs, such as alopecia areata, psoriasis, and rheumatoid arthritis, albeit the effect size was small. This finding could be associated with autoimmune flare-ups following repeated mRNA vaccination, which can cause subclinical diseases to become active and diagnosed[26-28]. The result of our study may indicate the necessity for additional monitoring when administering booster vaccinations. However, it should be interpreted cautiously due to the potential healthy vaccine effect. In addition, booster vaccinations have shown substantial safety and potential benefits of improving humoral immune response preventing COVID-19 diagnosis or reducing disease severity[29]. Moreover, an additional dose of the vaccine could serve as a strategy to address the limitation of its waning efficacy over time[30]. Therefore, our results are not sufficient to discourage booster vaccination and suggest that regular and long-term monitoring may be necessary to ensure the early detection and management of any emerging risks associated with repeated vaccinations.

This study has several strengths. First, we used the national medical data of ~10 million people and national information on COVID-19 infection and vaccination profiles. Second, the risk of incident AI-CTDs was measured with a large sample size and longer observation period, more than the mean follow-up period of 471 days, than that used in previous studies[18]. Third, we designed a historical control cohort to minimise selection bias and examined the reliability of the analysis by evaluating positive and negative outcome controls. Fourth, we considered several confounding factors such as sex, age, type of mRNA vaccine, cross-vaccination, and COVID-19 diagnosis status to consider their potential impacts and designed an analysis treating booster vaccination as a time-varying covariate to account for its variability during the observation.

However, this study has some limitations. First, the analysis was conducted on individuals belonging to a single ethnic group. Since autoimmune disease-associated single nucleotide polymorphisms vary by ethnicity, our results may not be generalisable to other populations[31]. Second, although our study has one of the longest follow-up periods among mRNA vaccine studies reported to date, this duration may still be considered too short, given that the development of AI-CTDs can take years to decades after trigger exposure[32]. Moreover, the observation period of 2 years before the index date may not have been long enough to identify pre-existing AI-CTDs due to their indolent onset. Therefore, some incident cases in this study could have had their onset prior to the observation. Third, considering the global decline in the use of healthcare services during the COVID-19 pandemic, some outcomes of interest may have been underdiagnosed during this period[33,34]. Nevertheless, we investigated negative control outcomes to address these concerns. Fourth, potential misclassified cases related to using ICD-10 code claim data could be existed in our study. To mitigate this risk, we considered patients with three or more visits under the same ICD-10 code as having AI-CTDs.

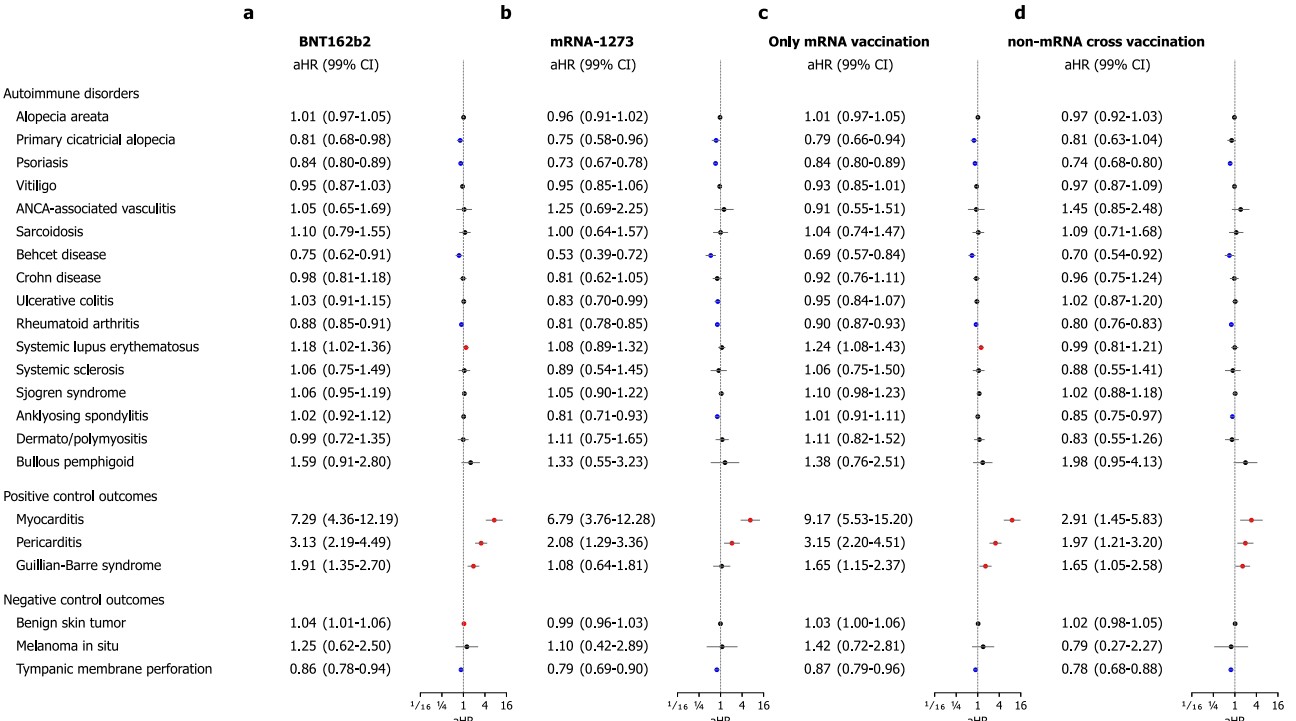

| | a | b | c | d |
|---|---|---|---|---|
| | BNT162b2 | mRNA-1273 | Only mRNA vaccination | non-mRNA cross vaccination |
| | aHR (99% CI) | aHR (99% CI) | aHR (99% CI) | aHR (99% CI) |
| **Autoimmune disorders** | | | | |
| Alopecia areata | 1.01 (0.97-1.05) | 0.96 (0.91-1.02) | 1.01 (0.97-1.05) | 0.97 (0.92-1.03) |
| Primary cicatricial alopecia | 0.81 (0.68-0.98) | 0.75 (0.58-0.96) | 0.79 (0.66-0.94) | 0.81 (0.63-1.04) |
| Psoriasis | 0.84 (0.80-0.89) | 0.73 (0.67-0.78) | 0.84 (0.80-0.89) | 0.74 (0.68-0.80) |
| Vitiligo | 0.95 (0.87-1.03) | 0.95 (0.85-1.06) | 0.93 (0.85-1.01) | 0.97 (0.87-1.09) |
| ANCA-associated vasculitis | 1.05 (0.65-1.69) | 1.25 (0.69-2.25) | 0.91 (0.55-1.51) | 1.45 (0.85-2.48) |
| Sarcoidosis | 1.10 (0.79-1.55) | 1.00 (0.64-1.57) | 1.04 (0.74-1.47) | 1.09 (0.71-1.68) |
| Behcet disease | 0.75 (0.62-0.91) | 0.53 (0.39-0.72) | 0.69 (0.57-0.84) | 0.70 (0.54-0.92) |
| Crohn disease | 0.98 (0.81-1.18) | 0.81 (0.62-1.05) | 0.92 (0.76-1.11) | 0.96 (0.75-1.24) |
| Ulcerative colitis | 1.03 (0.91-1.15) | 0.83 (0.70-0.99) | 0.95 (0.84-1.07) | 1.02 (0.87-1.20) |
| Rheumatoid arthritis | 0.88 (0.85-0.91) | 0.81 (0.78-0.85) | 0.90 (0.87-0.93) | 0.80 (0.76-0.83) |
| Systemic lupus erythematosus | 1.18 (1.02-1.36) | 1.08 (0.89-1.32) | 1.24 (1.08-1.43) | 0.99 (0.81-1.21) |
| Systemic sclerosis | 1.06 (0.75-1.49) | 0.89 (0.54-1.45) | 1.06 (0.75-1.50) | 0.88 (0.55-1.41) |
| Sjogren syndrome | 1.06 (0.95-1.19) | 1.05 (0.90-1.22) | 1.10 (0.98-1.23) | 1.02 (0.88-1.18) |
| Ankylosing spondylitis | 1.02 (0.92-1.12) | 0.81 (0.71-0.93) | 1.01 (0.91-1.11) | 0.85 (0.75-0.97) |
| Dermato/polymyositis | 0.99 (0.72-1.35) | 1.11 (0.75-1.65) | 1.11 (0.82-1.52) | 0.83 (0.55-1.26) |
| Bullous pemphigoid | 1.59 (0.91-2.80) | 1.33 (0.55-3.23) | 1.38 (0.76-2.51) | 1.98 (0.95-4.13) |
| **Positive control outcomes** | | | | |
| Myocarditis | 7.29 (4.36-12.19) | 6.79 (3.76-12.28) | 9.17 (5.53-15.20) | 2.91 (1.45-5.83) |
| Pericarditis | 3.13 (2.19-4.49) | 2.08 (1.29-3.36) | 3.15 (2.20-4.51) | 1.97 (1.21-3.20) |
| Guillian-Barre syndrome | 1.91 (1.35-2.70) | 1.08 (0.64-1.81) | 1.65 (1.15-2.37) | 1.65 (1.05-2.58) |
| **Negative control outcomes** | | | | |
| Benign skin tumor | 1.04 (1.01-1.06) | 0.99 (0.96-1.03) | 1.03 (1.00-1.06) | 1.02 (0.98-1.05) |
| Melanoma in situ | 1.25 (0.62-2.50) | 1.10 (0.42-2.89) | 1.42 (0.72-2.81) | 0.79 (0.27-2.27) |
| Tympanic membrane perforation | 0.86 (0.78-0.94) | 0.79 (0.69-0.90) | 0.87 (0.79-0.96) | 0.78 (0.68-0.88) |

**Fig. 5 | Stratified analyses of the incidence of autoimmune connective tissue disorders in the mRNA-based COVID-19 vaccination cohort compared with the historical control cohort according to the type of mRNA vaccine and the history of cross-vaccination with non-mRNA vaccination before the study index.** The forest plot depicts adjusted hazard ratios (aHRs) and 99% confidence intervals (CIs) in individuals with mRNA-based COVID-19 vaccination compared with historical controls. The point estimate (centre) represents the aHR, and the horizontal line (error bar) shows the range of the 99% CI. The incident risks of autoimmune disorder outcomes were stratified by the type of mRNA vaccine ((**a**) BNT162b2 or (**b**) mRNA-1273) and the history of cross-vaccination ((**c**) Only mRNA vaccination or

(**d**) Cross-vaccination with non-mRNA vaccination (AZD12222 or Ad26.COV2.S)). Additional details, including unadjusted HRs and population at risk, were provided in Supplementary Figures.; Subgroup who received the BNT162b2 vaccine (Supplementary Fig. 8), subgroup who received the mRNA-1273 vaccine (Supplementary Fig. 9), subgroup who received only mRNA-based vaccines (Supplementary Fig. 10), and subgroup who had a history of cross-vaccination with any non-mRNA vaccines (Supplementary Fig. 11). Abbreviations: aHR, adjusted hazard ratio; ANCA, antineutrophil cytoplasmic antibody; CI, confidence interval; COVID-19, coronavirus disease 2019.

In conclusion, our study results suggest that mRNA vaccination is generally not associated with a higher risk of most AI-CTDs. However, given that the risk of SLE and BP was increased in certain demographic conditions such as age and sex, long-term monitoring is necessary after mRNA vaccination for the development of AI-CTDs. Our results can provide clinical insights into mRNA therapeutics, and further research is needed regarding the association between mRNA-based vaccines and AI-CTDs[35].

## Methods
### Data source
This nationwide population-based cohort study was conducted using data from the KDCA COVID-19 NHIS (K-COV-N) cohort. The NHIS database provided comprehensive data consisting of demographics, insurance eligibility data (insurance type and area of residence), socioeconomic status (income level), inpatient and outpatient healthcare records (disease diagnoses and procedures), prescriptions, and national health examination results (alcohol use and smoking) of > 99% of the total population of South Korea[36]. The underlying disease of a population was confirmed when that disease was identified at ≥ 3 inpatient or outpatient visits using the corresponding International Classification of Diseases, Tenth Revision (ICD-10) diagnostic codes. The corresponding ICD-10 codes for the underlying diseases are summarised in Supplementary Table 2. Medical records that can identify past medical history before the index date and incident events during the observation period have been available since January 1, 2016. The South Korean government manages the NHIS COVID-19 registry, and the KDCA provides data regarding the COVID-19 vaccine,

such as the type, dose, and vaccination date. This study was approved by the Korean National Institute for Bioethics Policy, and the research number for this study was KDCA-NHIS-2023-1-500.

### Study population
In the NHIS database, 15,076,899 individuals, approximately 30% of the total South Korean population, underwent a general health examination in 2018. According to the data provider's regulations, we were required to limit our study population to 10 million due to privacy and data capacity constraints. Consequently, we randomly selected 9,945,450 participants, approximately 20% of the total population of South Korea. After excluding individuals with incomplete general health examination reports, 9,258,803 individuals vaccinated with at least one dose of the mRNA-based COVID-19 vaccine (BNT162b2 or mRNA-1273) until 31 December 2022 were selected. We subsequently extracted half of the primary cohort to establish the vaccination cohort, whose index was the date of administration of the first dose of the mRNA-based COVID-19 vaccine. As COVID-19 vaccination was conducted nationwide in South Korea, as of October 2022, the overall vaccination coverage rate among adults meeting the requirements for the primary series of each COVID-19 vaccine introduced in South Korea was 96.6%[13]. Therefore, using unvaccinated individuals as controls for comparison could have led to inappropriate cohort selection and potential selection bias. Instead, our study uses historical controls as comparators. The other half of the primary cohort was used to form historical control cohorts, while the observational period of the control group was shifted back by 2 years from the date of the first dose of mRNA vaccination of individuals. The two cohort groups were

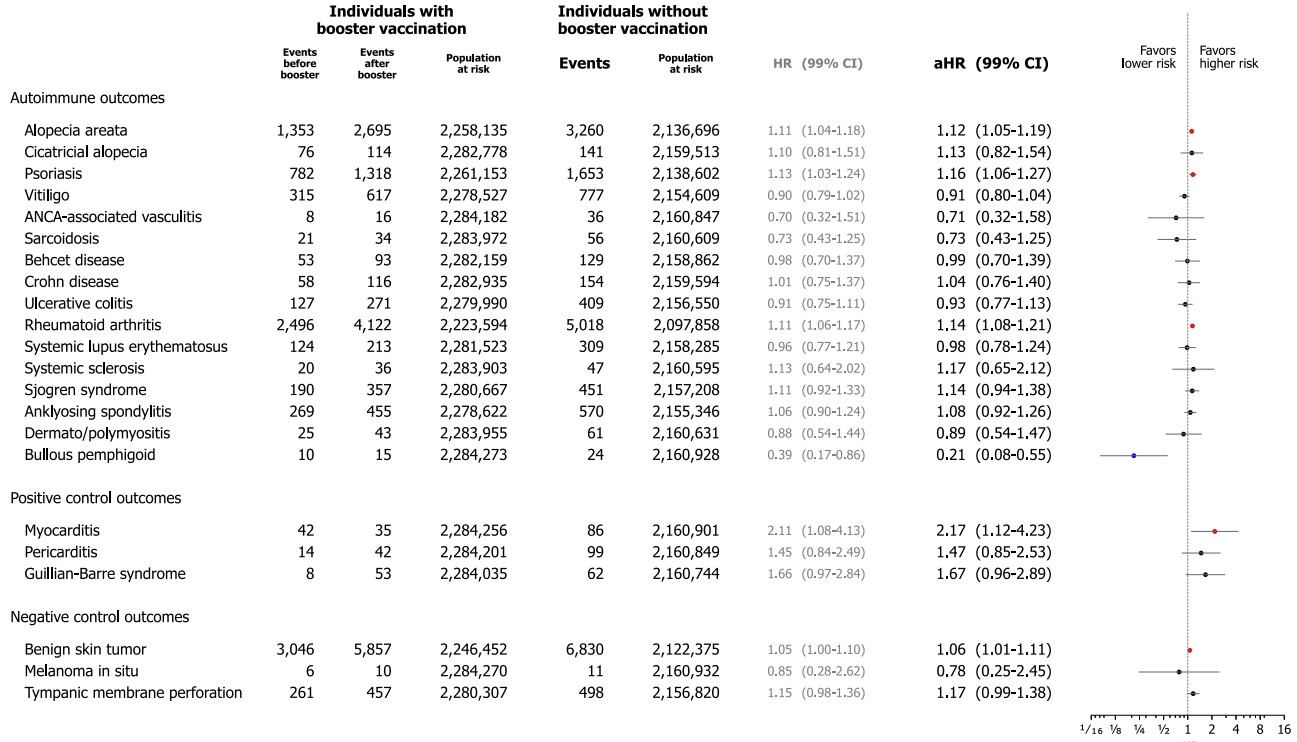

| | Individuals with booster vaccination | | | Individuals without booster vaccination | | | | | |
|---|---|---|---|---|---|---|---|---|---|
| | Events before booster | Events after booster | Population at risk | Events | Population at risk | HR (99% CI) | aHR (99% CI) | | Favors lower risk / Favors higher risk |
| **Autoimmune outcomes** | | | | | | | | | |
| Alopecia areata | 1,353 | 2,695 | 2,258,135 | 3,260 | 2,136,696 | 1.11 (1.04-1.18) | 1.12 (1.05-1.19) | | |
| Cicatricial alopecia | 76 | 114 | 2,282,778 | 141 | 2,159,513 | 1.10 (0.81-1.51) | 1.13 (0.82-1.54) | | |
| Psoriasis | 782 | 1,318 | 2,261,153 | 1,653 | 2,138,602 | 1.13 (1.03-1.24) | 1.16 (1.06-1.27) | | |
| Vitiligo | 315 | 617 | 2,278,527 | 777 | 2,154,609 | 0.90 (0.79-1.02) | 0.91 (0.80-1.04) | | |
| ANCA-associated vasculitis | 8 | 16 | 2,284,182 | 36 | 2,160,847 | 0.70 (0.32-1.51) | 0.71 (0.32-1.58) | | |
| Sarcoidosis | 21 | 34 | 2,283,972 | 56 | 2,160,609 | 0.73 (0.43-1.25) | 0.73 (0.43-1.25) | | |
| Behcet disease | 53 | 93 | 2,282,159 | 129 | 2,158,862 | 0.98 (0.70-1.37) | 0.99 (0.70-1.39) | | |
| Crohn disease | 58 | 116 | 2,282,935 | 154 | 2,159,594 | 1.01 (0.75-1.37) | 1.04 (0.76-1.40) | | |
| Ulcerative colitis | 127 | 271 | 2,279,990 | 409 | 2,156,550 | 0.91 (0.75-1.11) | 0.93 (0.77-1.13) | | |
| Rheumatoid arthritis | 2,496 | 4,122 | 2,223,594 | 5,018 | 2,097,858 | 1.11 (1.06-1.17) | 1.14 (1.08-1.21) | | |
| Systemic lupus erythematosus | 124 | 213 | 2,281,523 | 309 | 2,158,285 | 0.96 (0.77-1.21) | 0.98 (0.78-1.24) | | |
| Systemic sclerosis | 20 | 36 | 2,283,903 | 47 | 2,160,595 | 1.13 (0.64-2.02) | 1.17 (0.65-2.12) | | |
| Sjogren syndrome | 190 | 357 | 2,280,667 | 451 | 2,157,208 | 1.11 (0.92-1.33) | 1.14 (0.94-1.38) | | |
| Ankylosing spondylitis | 269 | 455 | 2,278,622 | 570 | 2,155,346 | 1.06 (0.90-1.24) | 1.08 (0.92-1.26) | | |
| Dermato/polymyositis | 25 | 43 | 2,283,955 | 61 | 2,160,631 | 0.88 (0.54-1.44) | 0.89 (0.54-1.47) | | |
| Bullous pemphigoid | 10 | 15 | 2,284,273 | 24 | 2,160,928 | 0.39 (0.17-0.86) | 0.21 (0.08-0.55) | | |
| **Positive control outcomes** | | | | | | | | | |
| Myocarditis | 42 | 35 | 2,284,256 | 86 | 2,160,901 | 2.11 (1.08-4.13) | 2.17 (1.12-4.23) | | |
| Pericarditis | 14 | 42 | 2,284,201 | 99 | 2,160,849 | 1.45 (0.84-2.49) | 1.47 (0.85-2.53) | | |
| Guillian-Barre syndrome | 8 | 53 | 2,284,035 | 62 | 2,160,744 | 1.66 (0.97-2.84) | 1.67 (0.96-2.89) | | |
| **Negative control outcomes** | | | | | | | | | |
| Benign skin tumor | 3,046 | 5,857 | 2,246,452 | 6,830 | 2,122,375 | 1.05 (1.00-1.10) | 1.06 (1.01-1.11) | | |
| Melanoma in situ | 6 | 10 | 2,284,270 | 11 | 2,160,932 | 0.85 (0.28-2.62) | 0.78 (0.25-2.45) | | |
| Tympanic membrane perforation | 261 | 457 | 2,280,307 | 498 | 2,156,820 | 1.15 (0.98-1.36) | 1.17 (0.99-1.38) | | |

1/16  1/8  1/4  1/2  1  2  4  8  16
aHR

**Fig. 6 | Risks of incident autoimmune connective tissue diseases within mRNA-based COVID-19 vaccination cohort according to prior history of booster vaccination.** The forest plot depicts adjusted hazard ratios (aHRs) with 99% confidence intervals (CIs) in individuals within the vaccination cohort according to prior history of booster vaccination, defined by the administration of 3rd additional dose of the mRNA-based COVID-19 vaccine following the completion of the two-dose primary series of the same mRNA-based COVID-19 vaccine. The point estimate (centre) represents the aHR, and the horizontal line (error bar) shows the range of the 99% CI. Among the vaccination cohort, 2,284,342 individuals were vaccinated with a booster dose and the extended Cox proportional hazard analyses treating booster vaccination as a time-varying covariate were conducted for the variability of vaccination status during the observation period. The numbers of events of autoimmune disorder outcomes and population at risk were presented for each group divided based on booster vaccination status, with the number of events specifically shown before and after the booster vaccination in the individuals with booster vaccination group. Abbreviations: aHR, adjusted hazard ratio; ANCA, antineutrophil cytoplasmic antibody; CI, confidence interval; COVID-19, coronavirus disease 2019; HR, hazard ratio.

followed up from the respective study index date to disease diagnosis, emigration, death, or the end of the study period. The vaccination group was observed until 31 December 2022, and the historical cohort group until 31 December 2020.

## Outcomes
We assessed the incidence and risk of developing AI-CTDs associated with the mRNA-based COVID-19 vaccine during the follow-up period by restricting the study population to patients without a history of the respective outcomes before the study index date. The occurrence of each outcome disease was defined when that disease was identified at ≥ 3 in- or outpatient visits using the corresponding ICD-10 diagnostic codes. We established and evaluated three positive control outcomes (myocarditis, pericarditis, and Guillain–Barre syndrome), which are reportedly significantly associated with the COVID-19 vaccine, as well as negative control outcomes (benign skin tumour, melanoma in situ, and tympanic membrane perforation), which are less likely to be associated with the COVID-19 vaccine, to ensure the validity of our study[37]. The corresponding ICD-10 codes of autoimmune connective tissue diseases and predefined positive and negative control outcomes are summarised in Supplementary Table 2.

## Covariates
Although both the vaccination and historical control cohorts were derived from the same primary cohort, there could be residual differences in baseline characteristics that were potentially linked to the occurrence of disease outcomes. Therefore, we considered predefined covariates, including demographics and socioeconomic status, such as age, sex, insurance type (standard vs Medicaid), income level (divided into quartiles based on health insurance premiums), area of residence (urban vs rural area), and general health examination data mentioned earlier. In the general health examination data, we established the current smoking status and defined drinking as routine alcohol consumption, regardless of the amount or frequency. In addition, we set several chronic diseases as predefined covariates and listed their corresponding ICD-10 codes in Supplementary Table 2. The covariates were balanced between the two cohorts using IPTW.

## Statistical analysis
The baseline demographic characteristics are presented as means with standard deviations and frequencies with percentages, depending on the variable types. Propensity scores for individuals were estimated based on predefined covariates representing the possibility of belonging to the vaccination cohort. These scores were used to calculate the inverse probability of treatment weights, obtained by dividing the probability of belonging to the vaccination cohort by 1 minus the probability of being in the vaccination cohort: the probability of belonging to the vaccination cohort / (1−the probability of being in the vaccination cohort). The covariate balance was assessed using standardised mean differences before and after the application of the IPTWs. Subsequently, the risk of predefined outcomes in the vaccination cohort was estimated and compared with that in the historical control cohort. Statistical analysis involved multivariable Cox proportional hazards analysis after adjusting for all predefined

covariates used to calculate the IPTWs. Each analysis for outcomes included only the population at risk by excluding those who had already been diagnosed with the target outcome at the index date or before. Stratified subgroup analyses were conducted according to the sex, age (< 40 vs ≥ 40), type of mRNA-based COVID-19 vaccine (BNT162b2 vs mRNA-1273), history of having a non-mRNA COVID vaccine such as the viral vector vaccine (ChAdOx1 nCoV-19 (AZD1222) or Ad26.COV2.S) prior to mRNA vaccination, and whether COVID-19 was diagnosed or not. We also aimed to further ascertain the impact of booster vaccination, defined by the administration of 3rd additional dose of the mRNA-based vaccine following the completion of the two-dose primary series of the same mRNA-based vaccine, on the incidence of AI-CTDs. As an individual's vaccination status may change during the observation, we conducted extended Cox proportional hazard analyses with booster vaccination as a time-varying covariate. To minimise the risk of false findings, such as type I errors due to multiple comparisons, we applied the Bonferroni correction for 27 predefined outcomes. Consequently, statistical significance was determined based on the adjusted confidence interval for 27 comparisons. All statistical analyses were conducted using SAS statistical software (version 9.4; SAS Institute, Cary, NC, USA) and R statistical software (version 3.4.1; R Foundation for Statistical Computing, Vienna, Austria).

### Reporting summary

Further information on research design is available in the Nature Portfolio Reporting Summary linked to this article.

## Data availability

The datasets analysed during the current study are available in the National Health Insurance Service in South Korea. This protects the confidentiality of the data and ensures that Information Governance is robust. Applications to access health data in South Korea should be submitted to the National Health Insurance Service in South Korea. Information can be found at https://nhiss.nhis.or.kr/bd/ab/bdaba000eng.do.

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

## Acknowledgements

This study used the KDCA and NHIS databases for policy and academic research. The research number for this study was KDCA-NHIS-2023-1-500. The KDCA is the Korea Disease Control and Prevention Agency, Republic of Korea. The NHIS is part of the National Health Insurance Service of the Republic of Korea. Funding: This research was supported by funding from the Research Programme of the Korea Medical Institute. This research was also supported by a grant from the Korea Health Technology Research and Development Project through the Korea Health Industry Development Institute (KHIDI), the Ministry of Health & Welfare, Republic of Korea (grant number: HI23C1506; S.L.), and by a grant from the National Research Foundation of Korea (NRF) grant funded by the Korea government (MSIT; no. RS-2023-00249120; S.L.).

## Author contributions

S.J.C. and S.L. have full access to all data in the study and take responsibility for the integrity and accuracy of the data analysis. S.W.J. and S.L. were involved in the conceptualisation and design of the study. S.W.J., J.J.J., Y.H.K., S.J.C. and S.L. contributed to the study methodology and data curation. S.W.J. and J.J.J. were involved in the analysis, data interpretation, and draughting of the manuscript. S.J.C. and S.L. contributed to the data validation, visualisation, and critical revision of the manuscript. S.W.J., J.J.J., Y.H.K., S.J.C. and S.L. administered the project.

## Competing interests

The authors declare no competing interests.
