## [Peer Review File · Nature Communications]

Long-Term Risk of Autoimmune Diseases After mRNA-based COVID-19 Vaccination: A Korean Population-based Nationwide Cohort StudyREVIEWER COMMENTS

Reviewer #1 (Remarks to the Author):

Thank you for the opportunity to review this manuscript. This important study, conducted in a large population, enhances our understanding of potential side effects associated with mRNA vaccines. The analytical approach is robust, and the inclusion of both positive and negative controls strengthens the findings. I have a few suggestions that could further enhance the manuscript's contribution to the literature.

Minor revisions

1. Introduction: The statement "More than 50% of the world's population has been diagnosed with it, according to a 2022 global seroprevalence survey" might be misleading. It would be more accurate to say that over 50% are estimated to have been infected, as per seroprevalence data, rather than formally diagnosed.
2. Introduction: The effectiveness of mRNA-based COVID-19 vaccines is presented as preventing 46–92% of SARS-CoV-2 infections, 74–87% of hospitalizations, and 62–92% of severe illnesses. This portrayal does not account for the decline in vaccine effectiveness over time or against variants such as Omicron. A more nuanced statement could reflect that vaccine effectiveness is initially high but wanes over time.
3. Methods: The rationale for dividing the full cohort into two parts is unclear, especially since using the entire cohort might allow each person to serve as their own control. Given the follow-up duration of less than 2 years, it may be beneficial to explain why the full cohort was not utilized.
4. Methods: With multiple outcomes assessed, statistically adjusting for multiple comparisons could help mitigate the risk of type I errors. This adjustment seems to be missing, and its inclusion could strengthen the study's findings.
5. Methods: The justification for using only 20% of the K-COV-N cohort does not seem to be provided. Clarification on this selection criterion would be helpful.
6. Methods: The methodology behind the subgroup analyses is not detailed. Specifying whether this involved stratified analysis, interaction tests, or another method would provide greater clarity.
7. Methods: It is unclear whether the covariate of booster vaccination was treated as a time-varying covariate. Clarification on this point is needed for understanding the analysis fully.
8. Methods: There seems to be a discrepancy regarding the exclusion of COVID-19 cases in the methods, as they appear to be included in some subgroup analyses. An explanation is needed to resolve this apparent inconsistency.
9. Results: Including unadjusted HRs in the figures alongside the adjusted HRs would offer useful insights into the impact of the adjustments made.
10. Results: The significant protective effects for vaccination noted for some outcomes may be influenced by the issue of multiple comparisons. It would be pertinent for the authors to address and discuss these findings.
11. Discussion: Although several effects are statistically significant, their clinical significance might be limited due to the small magnitude of these effects. Highlighting this aspect could provide a more balanced interpretation of the results.

Reviewer #1 (Remarks on code availability):

The analyses used standard commands. I do not think the code is relevant here.

Reviewer #2 (Remarks to the Author):

This paper describes the incidence of autoimmune connective tissue diseases after COVID-19 vaccination in a large population-based cohort from South Korea. The study sample size is large enough and there was adequate post-vaccination follow up time to detect many of these rare outcomes. As the authors note, South Korea had high uptake of the vaccine, so including concurrent unvaccinated controls would likely have introduced some selection biases into the analysis. For this reason, the authors elected to use a historical period of pre-pandemic time in a control cohort. Use of this historical control group could lead to biased findings if the rates of diagnoses for these conditions changed over time. Additionally, the outcomes were based on

diagnostic codes and these codes can not capture symptom onset and often are non-specific for the conditions of interest. Despite these limitations, this study does provide some information about the incidence of these autoimmune conditions following vaccination, which has been a concern raised by the public about the safety of these vaccines and has not been extensively evaluated.

Specific comments for the authors:

Page 3 – lines 49-50: The authors cite a seroprevalence study here, so would it be more accurate to say “infected with it” rather than “diagnosed with it”?

Page 8 – lines 171-176: “booster vaccination” – The authors’ use of “booster dose” is a little confusing. The primary entry criterion for both cohorts was the receipt of at least one dose of a mRNA COVID-19 vaccine. Since the primary series for the mRNA vaccines included two doses, does the authors’ use of “booster” refer to any subsequent dose (e.g., 2nd, 3rd, 4th) or does it truly refer to “boosters” given after primary series completion? This should be clarified. It is difficult to interpret the stratified analysis by “booster dose” and the conclusions drawn by the authors in this paragraph because the timing between doses and AI-CTD diagnosis is unclear. Perhaps the authors could have considered vaccination as a time-varying covariate in their models?

Page 9 – The authors used ICD codes to define autoimmune conditions in their analysis and these codes often misclassify cases. The potential for outcome misclassification should be stated as a limitation of this analysis. The authors did not use any supplemental data – laboratory tests, chart review – to validate the diagnosis codes. Additionally, many autoimmune conditions take years from onset to diagnosis, and it’s possible that many of the incident cases observed here had onset prior to vaccination and that should also be noted as a limitation.

Page 10 – line 220: “extracted half of the primary cohort to establish the vaccination cohort” – Was this selection done randomly?

Page 10 – line 222: “number of unvaccinated individuals was very low in South Korea” – What does “very low” mean? Could the authors provide an estimate of vaccination coverage here?

Page 11 – lines 227-228: “we randomly assigned the study index date of the historical control cohort by matching that of the vaccination cohort, but with a subtraction of 2 years” – I found this a little confusing. Do the authors mean that they used the control’s date of initial vaccination minus 2 years as the index date? Or did they take the index dates from the vaccination cohort minus 2 years and then randomly assign them to the controls?

Page 11 – lines 235-236: The authors restricted the study population to patients without a history of the respective outcomes before the index date – how far back did the authors look for pre-existing diagnoses? There are often years between symptom onset and diagnosis for many of these conditions. How many people were excluded for this reason and where is this shown in the CONSORT diagram?

Page 12 – lines 258-259: “propensity scores for individuals were estimated based on predefined covariates representing the possibility of belonging to the vaccination cohort” – I think the use of propensity scores and IPTW was perhaps not needed in this analysis. All of the participants included in the two cohorts were vaccinated. It is not clear how the two cohorts were selected, but the SMDs presented in Table 1 would suggest that it was done randomly. With the exception of age and a couple of comorbidities, the two cohorts were well-balanced before weighting, and the authors note that they adjusted their Cox models for all of the covariates used to calculate IPTWs.

Tables and Figures

Some of the figures seem to be included in both the main text and supplemental materials. Supplementary Fig 1 could be cut, since it does not add much information beyond what is shown in table form.

Reviewer comments	Author response	Page number in revised paper
Reviewer 1		
Thank you for the opportunity to review this manuscript. This important study, conducted in a large population, enhances our understanding of potential side effects associated with mRNA vaccines. The analytical approach is robust, and the inclusion of both positive and negative controls strengthens the findings. I have a few suggestions that could further enhance the manuscript's contribution to the literature. Minor revisions 1. Introduction: The statement "More than 50% of the world's population has been diagnosed with it, according to a 2022 global seroprevalence survey" might be misleading. It would be more accurate to say that over 50% are estimated to have been infected, as per seroprevalence data, rather than formally diagnosed.	We appreciate your detailed reviews and critical comments for improving our study. The below are our point-by-point responses for your comments. We agree with your correction. According to the reference #2, the cited study is a seroprevalence study with the objective of estimating the extent of infection and seropositivity to the virus. Therefore, it would be more appropriate to use 'infected' rather than 'diagnosed,' as you suggested. We have made the correction in the revised manuscript.	Main manuscript Page 3, lines 47-49
2. Introduction: The effectiveness of mRNA-based COVID-19 vaccines is presented as preventing 46–92% of SARS-CoV-2 infections, 74–87% of hospitalizations, and 62–92% of severe illnesses. This portrayal does not account for the decline in vaccine effectiveness over time or against variants such as Omicron. A more nuanced statement could reflect that vaccine effectiveness is initially high but wanes over time.	We agree with your comment that the effectiveness of vaccines wanes over time and in the presence of virus variants. We have modified the sentence to reflect this context: "In particular, though effectiveness of vaccine wanes over time and as virus variants such as Omicron, mRNA-based COVID-19 vaccines show generally significant efficacy, preventing 46–92% of SARS-CoV-2 infections, 74–87% of hospitalisations, and 62–92% of severe illnesses as defined by the National Institutes of Health criteria."	Main manuscript Page 3, lines 52-55
3. Methods: The rationale for dividing the full cohort into two parts is unclear, especially since using the entire cohort might allow each person to serve as their own control. Given the follow-up duration of less than 2 years, it may be beneficial to explain why the full cohort was not utilized.	We appreciate your concern regarding the division of the cohort. As our investigation centered on incident autoimmune diseases, it was imperative to accurately define the population at risk.	Not applicable

	If the entire cohort is utilized with each individual serving as their own control from a past time point (historical control), those diagnosed with the outcome of interest in the past cannot be included in the current population at risk. This is because their prior diagnosis disqualifies them from being at risk for a new onset of the same condition within the framework of the study. To resolve these methodological challenges, we segregated the complete cohort into two groups (a vaccinated group and a historical control group). This design has been adopted by some COVID-19 research as follows:  1. Xie, Y., Xu, E., Bowe, B. et al. Long-term cardiovascular outcomes of COVID-19. Nat Med 28, 583–590 (2022). https://doi.org/10.1038/s41591-022-01689-3 2. Xu, E., Xie, Y. & Al-Aly, Z. Long-term neurologic outcomes of COVID-19. Nat Med 28, 2406–2415 (2022). https://doi.org/10.1038/s41591-022-02001-z 3. Al-Aly, Z., Bowe, B. & Xie, Y. Long COVID after breakthrough SARS-CoV-2 infection. Nat Med 28, 1461–1467 (2022). https://doi.org/10.1038/s41591-022-01840-0 We hope this clarifies our rationale and methodological choices.	
4. Methods: With multiple outcomes assessed, statistically adjusting for multiple comparisons could help mitigate the risk of type I errors. This adjustment seems to be missing, and its inclusion could strengthen the study's findings.	We appreciate your detailed review for enhancing our study. We agree that multiple comparisons could increase the risk of type I error. To address this, we performed Bonferroni corrections for 27 predefined outcomes, resulting in adjusted 99.81% confidence intervals	Main manuscript Page 4, lines 91-93 Page 12, lines 285-287

	(CIs) for determining statistical significance. Consequently, we have decided to use the adjusted 99.81% CIs to determine significance in all analyses, but have presented them as 99% CIs for clarity and simplicity. Some of our results have changed after applying this correction, and we have revised our manuscript accordingly.	
5. Methods: The justification for using only 20% of the K-COV-N cohort does not seem to be provided. Clarification on this selection criterion would be helpful.	Thank you for your insightful comment to clarify our selection criteria. In 2018, approximately 15 million individuals (about 30% of the total population of South Korea) underwent general health examinations. According to the data regulation and policy for the K-COV-N cohort database, the NHIS cannot provide data for more than 10 million individuals due to privacy and capacity constraints. Therefore, we had to extract our study population from primary cohort. As a result, we randomly selected about 20% of the total population of South Korea (n=9,945,450) with pseudorandom number generator to form our study population. We have added the following sentence to the study population section of the methods: "In the NHIS database, 15,076,899 individuals, approximately 30% of the total South Korean population, underwent a general health examination in 2018. According to the data provider's regulations, we were required to limit our study population to 10 million due to privacy and data capacity constraint. Consequently, we randomly selected 9,945,450 participants, approximately 20% of the total population of South Korea."	Main manuscript Page 10, lines 225-228

6. Methods: The methodology behind the subgroup analyses is not detailed. Specifying whether this involved stratified analysis, interaction tests, or another method would provide greater clarity.	We appreciate your detailed review and the opportunity to improve our study's methodology. Most subgroup analyses (sex, age, type or mRNA vaccine, prior non-mRNA cross vaccination, and COVID-19 diagnosis status) in our study are stratified analyses. However, the reviewers' comment led us to review the methodology of our analysis, and we noticed that booster vaccination should be treated as a time-varying covariate. Therefore, we additionally performed extended Cox proportional hazard analysis with booster vaccination as time-varying covariates, and we have presented these revised results in the newly updated Figure 5. Additionally, we have added the following sentence to the statistical analysis section of the methods: "Stratified subgroup analyses were conducted according to the sex, age (<40 vs. ≥40), type of mRNA-based COVID-19 vaccine (BNT162b2 vs. mRNA-1273), history of having a non-mRNA COVID vaccine such as the viral vector vaccine (ChAdOx1 nCoV-19 (AZD1222) or Ad26.COV2.S) prior to mRNA vaccination, and whether COVID-19 was diagnosed or not. We also aimed to further ascertain the impact of booster vaccination, defined by the administration of 3rd additional dose of the mRNA-based vaccine following the completion of the two-dose primary series of the same mRNA-based vaccine, on the incidence of AI-CTDs. As an individual's vaccination status may change during the observation, we conducted extended	Main manuscript Pages 5-6, lines 115-137 Page 8, lines 170-180 Page 12, lines 278-285
--	--	--

	Cox proportional hazard analyses with booster vaccination as a time-varying covariate."	
7. Methods: It is unclear whether the covariate of booster vaccination was treated as a time-varying covariate. Clarification on this point is needed for understanding the analysis fully.	This comment seems to be related to your 6th comment and we appreciate your detailed review for clarifying our analysis. First, we have defined booster vaccination as the administration of 3rd additional dose of the mRNA-based COVID-19 vaccine following the completion of the two-dose primary series of the same mRNA-based COVID-19 vaccine. As you mentioned, we additionally performed extended Cox proportional hazard analysis with booster vaccination as a time-varying covariate, and modified all applicable contents with providing newly updated Figure 5. The reanalysis revealed that booster vaccination was associated with an increased risk of developing certain AI-CTDs, such as alopecia areata, psoriasis, and rheumatoid arthritis. This finding has been further discussed in the discussion section of revised manuscript.	Main manuscript Page 6, lines 132-137 Page 8, lines 170-180 Page 12, lines 281-285
8. Methods: There seems to be a discrepancy regarding the exclusion of COVID-19 cases in the methods, as they appear to be included in some subgroup analyses. An explanation is needed to resolve this apparent inconsistency.	We apologize for any confusion caused and appreciate your attention to this detail. There was indeed an error in the methods section regarding the exclusion of individuals diagnosed with COVID-19. In our study, our study did not exclude these individuals; all participants were included regardless of their COVID-19 diagnosis. We have addressed this inconsistency by removing the incorrect statement from the methods section in the revised manuscript.	Main manuscript Page 4, lines 75-77 (Erased) Page 10, line 231 (Erased) Page 17, line 420 (Erased)
9. Results: Including unadjusted HRs in the figures alongside the adjusted HRs would offer useful insights into the impact of the adjustments made.	Thank you for your valuable suggestion. According to your comment, we have revised the figures and added supplementary figures depicting both unadjusted and adjusted hazard ratios (HRs) for main and subgroup analyses.	Main manuscript Page 5, lines 120-122 Figure 2, figure 5, and supplementary figures 2-11

10. Results: The significant protective effects for vaccination noted for some outcomes may be influenced by the issue of multiple comparisons. It would be pertinent for the authors to address and discuss these findings.	We appreciate your detailed review and the opportunity to clarify and strengthen our study. This comment seems to be related to your 4th comment. We would like to clarify that our study does not primarily aim to highlight the protective effects of vaccination. Given the observational nature of the study and the use of historical control cohort, there are inherent biases preventing us from claiming any definitive protective effects of mRNA vaccination. We recognized the possibility of false findings, such as type I errors, due to multiple comparisons. To mitigate the risk of type I errors, we applied Bonferroni corrections for 27 predefined outcomes and used 99.81% confidence interval to determine statistical significance.	Main manuscript Page 4, lines 91-93 Page 12, lines 285-287
11. Discussion: Although several effects are statistically significant, their clinical significance might be limited due to the small magnitude of these effects. Highlighting this aspect could provide a more balanced interpretation of the results.	Thanks for your thoughtful comment. Following reviewers' feedbacks, we have adjusted the confidence intervals and reanalyzed the booster vaccination as time-varying covariate. Consequently, we found more significant results for several AI-CTDs in booster vaccination analyses. We have extensively revised the results and discussion section, with restructuring for the balanced interpretation. We were very cautious about the probable "protective effect" of mRNA vaccines on some AI-CTDs showing a significant decrease for several reasons. Firstly, patients who have completed the primary vaccination series are generally healthier individuals, which could	Main manuscript Page 8, lines 170-180

	contribute to the low incidence of some AI-CTDs. Additionally, claiming a protective effect of mRNA vaccines could mask the need for long-term surveillance of AI-CTDs that are currently observed to have decreased, considering the indolent course of AI-CTDs. Conversely, AI-CTDs that significantly increased in patients who completed the vaccination may be more meaningfully addressed as they represent AI-CTDs diagnosed despite the individuals being generally healthier. Therefore, we believed that there is more bias towards the “protective effect” of mRNA vaccines on diseases showing a decrease in incidence compared to the "immunologic trigger effect" of mRNA vaccines on diseases showing an increase in incidence. Hence, we approached the interpretation of diseases showing a decrease with caution regarding claiming a protective effect in our study.	
(Remarks on code availability): The analyses used standard commands. I do not think the code is relevant here.	As you commented, we have added the following sentence to the code availability status after the methods: “This study did not generate new or customized code/algorithm. Statistical analyses were performed using SAS (version 9.4; SAS Institute, Cary, NC, USA) for analysis of big data. The codes utilized in this analysis are available from the corresponding author.”	Main manuscript Page 14, lines 298-301

Reviewer 2		
This paper describes the incidence of autoimmune connective tissue diseases after COVID-19 vaccination in a large population-based cohort from South Korea. The study sample size is large enough and there was adequate post-vaccination follow up time to detect many of these rare outcomes. As the authors note, South Korea had high uptake of the vaccine, so including concurrent unvaccinated controls would likely have introduced some selection biases into the analysis. For this reason, the authors elected to use a historical period of pre-pandemic time in a control cohort. Use of this historical control group could lead to biased findings if the rates of diagnoses for these conditions changed over time. Additionally, the outcomes were based on diagnostic codes and these codes can not capture symptom onset and often are non-specific for the conditions of interest. Despite these limitations, this study does provide some information about the incidence of these autoimmune conditions following vaccination, which has been a concern raised by the public about the safety of these vaccines and has not been extensively evaluated. Specific comments for the authors: Page 3 – lines 49-50: The authors cite a seroprevalence study here, so would it be more accurate to say “infected with it” rather than “diagnosed with it”?	We appreciate your detailed reviews and critical comments for improving our study. The below are our point-by-point responses for your comments. We agree with your correction. According to the reference #2, the cited study is a seroprevalence study with the objective of estimating the extent of infection and seropositivity to the virus. Therefore, it would be more appropriate to use 'infected' rather than 'diagnosed,' as you suggested. We have made the correction in the revised manuscript.	Main manuscript Page 3, lines 47-49
Page 8 – lines 171-176: “booster vaccination” – The authors’ use of “booster dose” is a little confusing. The primary entry criterion for both cohorts was the receipt of at least one dose of a mRNA COVID-19 vaccine. Since the primary series for the mRNA vaccines included two doses, does the authors’ use of “booster” refer to any subsequent dose (e.g., 2nd, 3rd, 4th) or does it truly refer to “boosters” given after primary series completion? This should be clarified. It is difficult to interpret the stratified analysis by “booster dose” and the conclusions drawn by the authors in this paragraph because the timing between	Thank you for your insightful review and for highlighting the need for clarification regarding our use of the term "booster vaccination". We acknowledge that our manuscript should clearly define the term "booster vaccination." Both two mRNA-based vaccines (BNT162b2 and mRNA-1273) are administered as a series of two doses, and booster vaccination typically refers to any additional dose administered after the completion of this primary series, based on FDA	Main manuscript Page 6, lines 132-137 Page 8, lines 170-180 Page 12, lines 281-285

doses and AI-CTD diagnosis is unclear. Perhaps the authors could have considered vaccination as a time-varying covariate in their models?

authorization. In our manuscript, we use the term "Booster vaccination" to denote the administration of a third or subsequent dose following the two-dose primary series. We recognize that this may have caused confusion, and we have revised the methods section of our manuscript to include the following definition: "We also aimed to further ascertain the impact of booster vaccination, defined by the administration of 3rd additional dose of the mRNA-based vaccine following the completion of the two-dose primary series of the same mRNA-based vaccine, on the incidence of AI-CTDs." To support this definition, we have included the following references in our revised manuscript:

1. World Health Organization. Living guidance for clinical management of COVID-19. November 23, 2021 (<https://www.who.int/publications/i/item/WHO-2019-nCoV-clinical-2021-2>).
2. Barda N, Dagan N, Cohen C, et al. Effectiveness of a third dose of the BNT162b2 mRNA COVID-19 vaccine for preventing severe outcomes in Israel: an observational study. *Lancet*. 2021;398(10316):2093-2100. doi:10.1016/S0140-6736(21)02249-2
3. Abu-Raddad LJ, Chemaitelly H, Ayoub HH, et al. Effect of mRNA Vaccine Boosters against SARS-CoV-2 Omicron Infection in Qatar. *N Engl J Med*. 2022;386(19):1804-1816. doi:10.1056/NEJMoa2200797
4. Arbel R, Sergienko R, Hammerman A. BNT162b2 Vaccine Booster and Covid-19 Mortality. Reply. *N Engl J Med*. 2022;386(10):1000-1001. doi:10.1056/NEJMc2120044

5. Eliakim-Raz N, Leibovici-Weisman Y, Stemmer A, et al. Antibody Titers Before and After a Third Dose of the SARS-CoV-2 BNT162b2 Vaccine in Adults Aged ≥ 60 Years. JAMA. 2021;326(21):2203-2204. doi:10.1001/jama.2021.19885

Additionally, your suggestion prompted us to review our analytical methodology. We additionally performed extended Cox proportional hazard analysis with booster vaccination as a time-varying covariate.

It is determined that a total of 2,284,342 individuals within mRNA vaccination cohort had received the booster vaccination. The reanalysis revealed that the incident risk of some AI-CTDs was higher in individuals with booster vaccination. We have presented these revised results in the newly updated Figure 5, incorporating all necessary adjustments. Additionally, this finding has been further discussed in the discussion section of revised manuscript and we have added the following sentence to the statistical analysis section of the methods: "Stratified subgroup analyses were conducted according to the sex, age (<40 vs. ≥ 40), type of mRNA-based COVID-19 vaccine (BNT162b2 vs. mRNA-1273), history of having a non-mRNA COVID vaccine such as the viral vector vaccine (ChAdOx1 nCoV-19 (AZD1222) or Ad26.COV2.S) prior to mRNA vaccination, and whether COVID-19 was diagnosed or not. We also aimed to further ascertain the impact of booster vaccination, defined by the administration of 3rd additional dose of the mRNA-based vaccine following the completion of

	the two-dose primary series of the same mRNA-based vaccine, on the incidence of AI-CTDs. As an individual's vaccination status may change during the observation, we conducted extended Cox proportional hazard analyses with booster vaccination as a time-varying covariate."	
Page 9 – The authors used ICD codes to define autoimmune conditions in their analysis and these codes often misclassify cases. The potential for outcome misclassification should be stated as a limitation of this analysis. The authors did not use any supplemental data – laboratory tests, chart review – to validate the diagnosis codes. Additionally, many autoimmune conditions take years from onset to diagnosis, and it's possible that many of the incident cases observed here had onset prior to vaccination and that should also be noted as a limitation.	Thank you for your critical comments. We acknowledged the potential misclassified issues related to using ICD-10 codes claim data in our study, as you mentioned, and we tried to mitigate these issues. The overall positive predictive value (PPV) of claim data versus actual diagnoses in the Korean Health Insurance database is reported to be approximately 82% according to following reference: - Park EJS, Jeon S, Lee S, Lee J, Choi D. Report of the evaluation for validity of discharged diagnoses in Korean Health Insurance database. Health Insurance Review and Assessment Service; 2017 To reduce the risk of underestimation or overestimation, we considered patients with three or more visits under the corresponding ICD code as having incident AI-CTDs. Furthermore, while it would provide more valid data to correlate clinical symptoms and various laboratory tests, the nature of claims databases limits access to such specific medical data. Despite these measures, we acknowledge that errors may still exist. In response to your suggestions, we have incorporated these points into the limitations section of our discussion.	Main manuscript Pages 8-9, lines 191-196 Page 9, lines 198-200

	Additionally, we were able to examine medical records from 1 January 2016 onward due to the capacity constraints of our database, the K-COV-N cohort database. We acknowledged that the observation period of less than two years may not have been sufficient to identify pre-existing diagnoses of AI-CTDs due to their indolent onset and slow development. As a result, some cases identified during our study may have originated before the administration of the vaccine. We have addressed this concern by enhancing and elaborating on these points within the limitations section of our revised manuscript.	
Page 10 – line 220: “extracted half of the primary cohort to establish the vaccination cohort” – Was this selection done randomly?	The selection of half of the primary cohort to establish the vaccination cohort was performed using a pseudorandom number generator. Each individual in the primary cohort was assigned a number, either 1 or 2, based on this generator, with no additional selection criteria involved. This method ensured that the selection was random.	Not applicable
Page 10 – line 222: “number of unvaccinated individuals was very low in South Korea” – What does “very low” mean? Could the authors provide an estimate of vaccination coverage here?	Based on your comment, we would like to provide further clarification regarding the term “very low” in the context of unvaccinated individuals in South Korea. As our study’s primary cohort consisted of individuals who had received at least one dose of a vaccine by 2022, we aimed to confirm the vaccination coverage for that period. Since KDCA provided official reports up to October 2022, we referenced the vaccination coverage as of 28 October 2022. In South Korea, as of October 2022, the vaccination coverage rate among adults who met the requirements for the primary series of each vaccine was 96.6%. We thought that this high rate suggests that the majority of the South	Main manuscript Page 4, lines 77-78 Pages 10-11, lines 232-235

	Korean population was vaccinated, accounting for those who might not be vaccinated due to various factors such as age, geographical location or underlying medical conditions. As you commented, we have revised the manuscript to replace the ambiguous term "very low" with the precise figure "96.6%". The methods section now includes the following revised sentence for clarity: "As COVID-19 vaccination was conducted nationwide in South Korea, as of October 2022, the overall vaccination coverage rate among adults meeting the requirements for primary series of each COVID-19 vaccine introduced in South Korea was 96.6%." Additionally, we have provided the URL to the KDCA press release as a reference (Newly updated reference #13) for this statistic: https://www.kdca.go.kr/board/board.es?mid=a20501010000&bid=0015&list_no=720988&cg_code=&act=view&nPage=1&newsField=202210	
Page 11 – lines 227-228: “we randomly assigned the study index date of the historical control cohort by matching that of the vaccination cohort, but with a subtraction of 2 years” – I found this a little confusing. Do the authors mean that they used the control’s date of initial vaccination minus 2 years as the index date? Or did they take the index dates from the vaccination cohort minus 2 years and then randomly assign them to the controls?	We apologize for any confusion caused by our initial explanation and appreciate your attention to detail. The study index date for the historical control cohort was determined by taking the date of the first dose of mRNA vaccination for individual in the historical control cohort and subtracting 2 years (730 days). We have revised all applicable sections, including Figure 1, to accurately reflect this methodology.	Main manuscript Page 4, lines 79-81 Page 11, lines 237-239 Page 17, lines 413-415
Page 11 – lines 235-236: The authors restricted the study population to patients without a history of the respective outcomes before the index date – how far back did the authors look for pre-existing diagnoses? There are often years between symptom onset and diagnosis for many	Thank you for your detailed comments and for the opportunity to clarify our methodology. Regarding your query about the restriction of the study population to patients without a history of the respective outcomes before the index date,	Main manuscript Pages 8-9, lines 192-197 Page 10, lines 218-220 Page 17, lines 422-424

of these conditions. How many people were excluded for this reason and where is this shown in the CONSORT diagram?	we conducted our analysis by examining medical records from 1 January 2016 onward, as our study database, the K-COV-N cohort database, is available from that date due to database capacity constraints. In response to your comments, we have added these points into the data source section of our methods. We also recognized that an observation period of less than 2 years may not have been sufficient to identify pre-existing diagnoses of autoimmune connective tissue diseases (AI-CTDs) due to their indolent onset and slow development. This limitation has been addressed in the revised manuscript: "Second, although our study has one of the longest follow-up periods among mRNA vaccine studies reported to date, this duration may still be considered too short given that the development of AI-CTDs can take years to decades after trigger exposure.³² Moreover, the observation period of 2 years before the index date may not have been long enough to identify pre-existing AI-CTDs due to their indolent onset. Therefore, some incident cases in this study could have had their onset prior to the observation." Furthermore, we have acknowledged the importance of analyzing populations at risk. As our study is not a randomized controlled trial, we did not use a CONSORT diagram. Instead, we have added population at risk for each predefined outcomes to the revised figures (Figures 2 and 5) and supplementary figures (Supplementary figure 2-11).	
---	--	--

Page 12 – lines 258-259: “propensity scores for individuals were estimated based on predefined covariates representing the possibility of belonging to the vaccination cohort” – I think the use of propensity scores and IPTW was perhaps not needed in this analysis. All of the participants included in the two cohorts were vaccinated. It is not clear how the two cohorts were selected, but the SMDs presented in Table 1 would suggest that it was done randomly. With the exception of age and a couple of comorbidities, the two cohorts were well-balanced before weighting, and the authors note that they adjusted their Cox models for all of the covariates used to calculate IPTWs.

We appreciate your detailed review and valuable comments.

First, as you pointed out, both two cohorts were randomly extracted from the same primary cohort and were balanced even before applying inverse probability treatment weighting (IPTW). As a result, the standardized mean differences SMDs were not substantial before weighting. However, we postulated that there could be residual differences in baseline characteristics and medical history of chronic diseases potentially linked to the occurrence of disease outcomes between two groups. Therefore, we have decided applying IPTW and found that applying IPTW further reduced the SMDs. Previous studies for COVID-19 which employed histological controls have employed similar methodologies with propensity score matching and IPTW. The following references illustrate similar approaches:

- Xie, Y., Xu, E., Bowe, B. et al. Long-term cardiovascular outcomes of COVID-19. *Nat Med* 28, 583–590 (2022). <https://doi.org/10.1038/s41591-022-01689-3>

- Xu, E., Xie, Y. & Al-Aly, Z. Long-term neurologic outcomes of COVID-19. *Nat Med* 28, 2406–2415 (2022). <https://doi.org/10.1038/s41591-022-02001-z>

- Al-Aly, Z., Bowe, B. & Xie, Y. Long COVID after breakthrough SARS-CoV-2 infection. *Nat Med* 28,

Main manuscript
Page 5, lines 120-122
Page 12, lines 268-269

	1461–1467 (2022). https://doi.org/10.1038/s41591-022-01840-0 Additionally, it was essential to consider that the population at risk for both groups differed for each outcome. Despite applying IPTW to minimize SMDs, we recognized that excluding individuals with prior diagnoses respectively from each cohort could introduce additional imbalances between two cohorts. Consequently, we believe that further adjusting for these potential imbalances was necessary. To address this, our Cox proportional hazard models adjusted the covariates which had been already used for IPTW. Therefore, to provide a comprehensive view of the impact of these adjustments, we added unadjusted HR and adjusted HR to the revised figures and supplementary figures, with the information on the population at risk for each outcome.	
Tables and Figures Some of the figures seem to be included in both the main text and supplemental materials. Supplementary Fig 1 could be cut, since it does not add much information beyond what is shown in table form.	We agree with your comment. As the information could be comprehensively understood from Table 1, we have removed the Supplementary Figure 1.	Not applicable (Erased)

REVIEWERS' COMMENTS

Reviewer #1 (Remarks to the Author):

The authors addressed my comments substantially and meaningfully. Thank you for considering them.

Reviewer #2 (Remarks to the Author):

Thank you for considering my review comments and addressing them thoroughly. I like the addition of the time-varying vaccination analysis.

Reviewers' comments	Author response	Page number in revised paper
Reviewer #1 (Remarks to the Author)		
The authors addressed my comments substantially and meaningfully. Thank you for considering them.	Our study has been significantly improved through your detailed reviews and constructive feedback. We are grateful and honoured to have gone through the revision process with you.	Not applicable
Reviewer #2 (Remarks to the Author)		
Thank you for considering my review comments and addressing them thoroughly. I like the addition of the time-varying vaccination analysis.	Our study has been significantly improved through your thoughtful comments and constructive feedback. We are grateful and honoured to have gone through the revision process with you.	Not applicable